# Segmented School Physical Activity and Weight Status in Children: Application of Compositional Data Analysis

**DOI:** 10.3390/ijerph18063243

**Published:** 2021-03-21

**Authors:** Ryan Donald Burns, Timothy A. Brusseau, Yang Bai, Wonwoo Byun

**Affiliations:** Department of Health & Kinesiology, University of Utah, Salt Lake City, UT 84112, USA; tim.brusseau@utah.edu (T.A.B.); Yang.Bai@utah.edu (Y.B.); won.byun@utah.edu (W.B.)

**Keywords:** body mass index, classroom, pedometer, physical education, recess

## Abstract

The purpose of this study was to apply compositional data analysis (CoDA) for the analysis of segmented school step counts and associate the school step count composition to body mass index (BMI) z-scores in a sample of children. Participants were 855 (51.8% female) children recruited from the fourth and fifth grades from four schools following a 7-h school schedule. Using piezoelectric pedometers, step count data were collected during physical education, recess, lunch, and during academic class time. A multi-level mixed effects model associated the step count composition with BMI z-scores. Compositional isotemporal substitution determined changes in BMI z-scores per reallocation of steps between pairs of school segments. A higher percentage of steps accrued during physical education (b = −0.34, 95%CI: −0.65–−0.03, *p* = 0.036) and recess (b = −0.47, 95%CI: −0.83–−0.11, *p* = 0.012), relative to other segments, was associated with lower BMI z-scores. Specifically, a 5% to 15% reallocation of steps accrued during lunchtime to either physical education or recess was associated with lower BMI z-scores, ranging from −0.07 to −0.25 standard deviation units. Focusing school-based promotion of physical activity during physical education and recess may have greater relative importance if targeted outcomes are weight-related.

## 1. Introduction

Physical activity is a key component of the energy balance equation [1,2,3]. The school environment has been a target to administer physical activity interventions to promote behavior change and enhance health within the pediatric population [4]. Single and multi-component interventions such as Comprehensive School Physical Activity Programs have been proposed and tested on their effectiveness to improve physical activity behaviors and health. Interventions tested using quasi-experimental research designs have shown that program implementation is associated with improved physical activity behaviors, fitness, and health outcomes [5,6,7]. However, when these programs are tested using randomized control trials, only small to moderate pooled effect sizes are observed [8,9], and some randomized control trials have shown no significant effect for school-based physical activity interventions to improve objective physical activity behaviors [10].

Possible explanations for these findings include the Structured Days Hypothesis, postulating that on days when children’s schedules are structured (e.g., school schedules), there is greater opportunity to engage in healthy behaviors compared to unstructured days (e.g., summer break) [11]. Thus, when comparing schools with school-based physical activity programs implemented to control schools without programming, the fact that both experimental arms include structured schedules may yield null or small effect findings because all schools contain health behavior-promoting environments [11]. Other explanations for a lack of conclusive effectiveness using school-based approaches to improve physical activity include low “buy-in” from schoolteachers and staff and barriers to expand, extend, and/or enhance new and existing physical activity opportunities in school settings, including quality physical education, which is a central tenet of many school-based physical activity programs and a school segment needed for facilitating lifetime physical activity engagement [12,13,14]. Due to the lack of strong evidence for coordinated school physical activity programming to be more effective than no programming for physical activity promotion [15], further investigation is warranted to determine how specific school segments such as physical education, recess, lunchtime, and the academic classroom can be better targeted for future interventions. To do this, determining how children accumulate physical activity during these specific segments and how accrued segmented physical activity is associated with health outcomes can provide important information.

Health behavior surveillance research within the psychomotor domain can involve a “Whole Day Matters” approach where the movement-based behaviors of physical activity, sedentary behavior, and sleep are analyzed concurrently because of their integrated and co-dependent nature across a 24-h day [16,17]. Novel analytic approaches for physical activity surveillance such as Compositional Data Analysis (CoDA) make it possible to account for the constrained nature of movement behaviors (i.e., physical activity, sedentary behavior, sleep) during a 24-h day [18,19]. CoDA also allows for the analysis of how reallocation of behaviors from one compositional part to another is associated with changes in health outcomes [18,19]. CoDA has been applied to accelerometer data with behavioral compositions being associated with weight status, fitness, gross motor skills, and cognitive outcomes [20,21,22,23,24]. Although CoDA’s application to accelerometer data is appropriate given the types of research questions in physical activity surveillance research, its application to data using other physical activity surveillance instruments, such as pedometers, may also have merit. CoDA can be applied to any data structure where compositional parts add to a whole (i.e., 100%) [18,19,22]. This concept can be generalized to the accumulation of step counts across a school day where step counts accrued during different school segments, such as during physical education and recess, integrate to total school day steps. Although analysis of segmented school steps is not time-use in nature, there may still be a degree of co-dependency between step counts accrued during different school segments that should be accounted for. Use of CoDA can account for this co-dependency between school segments and also allows for the analysis of how a hypothetical reallocation of steps between school segments is associated with changes in health markers. Additionally, the specific school segments of physical education and recess are being designed out of the school day in many countries but are arguably the most important segments for physical activity promotion in youth. It is important to determine how physical activity accrued during these segments relates to health outcomes relative to other non-physical activity-promoting segments to provide evidence of just how important physical education and recess are for health promotion within the pediatric population. Given the need to further examine how children accumulate physical activity during the school day and the advantages of CoDA for the analysis of compositional physical activity data, the purpose of this study was to apply CoDA for the analysis of segmented school step counts and associate the school step count composition to BMI z-scores in a sample of children.

## 2. Materials and Methods

### 2.1. Participants

Participants were a non-probability convenience sample of students recruited from 4 elementary schools located within the same school district from the Western United States. Descriptive statistics of the sample are reported within Table 1. The total sample included 855 students (n = 443 girls, n = 412 boys) from the fourth and fifth grades, who had a mean age of 9.7 ± 1.0 years old and were characterized by the following race/ethnicity backgrounds: White (n = 389; 45.5%), Black or African American (n = 80; 9.4%), Asian (n = 29; 3.4%), Hispanic-White (n = 310; 36.3%), Hawaiian/Pacific Islander (n = 42; 4.9%), and 5 (0.6%) students who did not respond. Given information from the school district website, approximately 60% of the sample was characterized as being from low-income families and received free and reduced cost lunch. Individual-level socioeconomic information was not obtained. Students were not compensated for participating in this research study and data collection was not permissible for the younger grade levels at the recruited schools. Written assent was obtained from the students and written consent was obtained from the parents before data collection. The University Institutional Review Board approved the protocols used in this study (IRB_00078226).

### 2.2. Instrumentation

BMI was calculated using standard procedures, taking a student’s weight in kilograms divided by the square or his or her height in meters. Height was measured to the nearest 0.01 m using a portable stadiometer (Seca 213; Hanover, MD, USA) and weight was measured to the nearest 0.1 kg using a portable medical scale (BD-590; Tokyo, Japan). Height and weight were collected individually in a private room during each student’s physical education class. The children wore light clothing and had their shoes off during the weight measurement. BMI z-scores were calculated by assessing the deviation of each child’s BMI value from the population mean BMI values reported in the BMI-for-age 2000 Centers for Disease Control and Prevention (CDC) growth chart using Stata’s “zanthro” package [25].

The New Lifestyles NL-1000 piezoelectric pedometer (New-Lifestyles Inc., Lee’s Summit, MO, USA) was used to assess step counts during different segments of school days. This specific pedometer has been validated for assessing physical activity in children and adolescents [26,27]. The pedometers were worn on the right side of the body at the level of the iliac crest above the right knee cap. All pedometers were provided an identification number and assigned to a student participant with a corresponding identification number. The pedometers were worn for 5 consecutive school days during school hours only.

### 2.3. Procedures

Each school had two 45-min physical education classes per week that were taught by a physical education specialist, an unstructured but supervised 15-min recess per day that provided some play equipment, and a 40-min lunch break per day that also incorporated a 20-min free play option after 20 min of eating, supervised by a faculty member. The physical education classes used a dynamic physical education model incorporating warm-up time, fitness time, skill development, and a cooperative game. Recess games tended to include soccer, tag games, use of playground equipment, and also sedentary behaviors (e.g., sitting and standing) and socializing. Each school had a 7-h school day. Both weight status and step counts were collected during the same week at each school during the Fall semester. Graduate research assistants distributed the pedometers prior to the start of the school day. The research assistants prompted students to record their step counts on a daily activity log before and after each class segment. The research assistants collected the pedometers from each student at the end of the school day and recorded the total school day step counts. Step counts were recorded for steps accumulated during physical education, steps during recess, and steps during lunch by subtracting the steps values before the segment from the post segment step value. Steps during the students’ academic classes were calculated by subtracting the combined steps accumulated during physical education, recess, and lunchtime from the daily total. This process was repeated for 5 consecutive school days. The average step count per school segment per student was used for analysis for the two days when all segment data were included (i.e., days with physical education). The percent averages for each school day segment were summed so that the school segments integrated into a school day total step count score (i.e., 100% of school steps). There are no recommendations on what constitutes an outlier or extreme step count score during specific school segments. Therefore, we used best judgement and determined that all scores in the dataset were plausible recordings. Daily missing data (<10%) were not used in the average school segment calculations and there were no missing average school step calculations. Specific data processing and CoDA procedures have been adopted from other work [18,19,20,21,22], with details provided within the Appendix A.

### 2.4. Statistical Analysis

Arithmetic and geometric means for step counts accrued during the four school segments were reported. Compositional mean bar plots were created to display the log ratio of geometric means for each sex to the mean of the whole sample. A compositional variation matrix was derived to communicate the variation of the calculated pair-wise log ratios. A variation coefficient close to zero indicates that there is higher co-dependency between two respective school segment step counts [18,19,20,21,22]. Higher co-dependency suggests that there is smaller variability in the log ratio between two school segments within the sample.

To examine the relative relationships among school steps accrued during physical education, recess, lunch, and during the academic class time with BMI z-scores, general linear regression models were employed. Sequential permutation was used to calculate parameter estimates for each of the four school day segments. Specifically, isometric log ratios (*ilrs*) expressed the ratio of percentage of steps accrued during one school segment relative to other school segments (e.g., school step activity during physical education to all other non-PE school day segments). Additional *ilrs* were calculated by permutating school segments in a sequential manner (see Appendix A). Three *ilrs* were included in the linear regression analyses; however, inferences about the primary contrast of interest (i.e., physical education relative to the three other school segments) were based on the first *ilr* pivot coordinate for each segment. The crude model consisted of utilizing a least-squares linear regression model using *ilr* predictors with no additional covariates. The adjusted model accounted for clustering of students within classrooms and the clustering of classrooms within schools using a multi-level mixed effects model with random intercepts. The adjusted model also accounted for race/ethnicity. Age and sex were not included as covariates within the adjusted model as they were used in the calculation of the BMI z-scores and their inclusion in the model would be redundant. Parameter estimates (b-coefficients) with corresponding 95% Confidence Intervals were reported.

Compositional isotemporal substitution was used to determine how reallocation of accrued steps between school segments was associated with changes in BMI z-scores [19,24,28]. Mean percentages for each school segment were used to predict BMI z-scores from which change values could be calculated based on a new step count composition (e.g., reallocating approximately 5% or 250 steps accrued during lunchtime to physical education). Technical details of this specific CoDA calculation have been described elsewhere [19,20]. All analyses had a statistical significance level set at *p* < 0.05 and were carried out using the Stata v15.0 statistical software package (StataCorp LLC, College Station, TX, USA).

## 3. Results

### Descriptive Statistics

CoDA descriptive statistics are reported within the Appendix A. Appendix A communicates the Arithmetic and Geometric means for step counts accrued during each school segment. The majority of the school day steps were accrued during physical education, followed closely by steps accrued during lunchtime. Steps accrued during academic class time made up the smallest percentage of total school day steps, followed by percentage of steps accrued during recess. Appendix A communicates the compositional variation matrix. Coefficients within the variation matrix ranged from 0.14 to 1.39. The highest coefficient was the log ratio variance between physical education and academic class time, indicating a relatively higher degree of independence between steps accrued during those two school segments. The smallest coefficient was the log ratio variance between recess and lunchtime followed closely by the log ratio variance between physical education and lunchtime, indicating a relatively high degree of co-dependency between these school segments. Descriptive differences comparing step count geometric means between girls and boys are presented in Appendix A. Boys accrued a relatively lower percentage of total school steps during physical education compared to girls. Conversely, boys accrued a relatively higher percentage of total school steps during all other school segments compared to girls.

Table 2 presents the unadjusted and adjusted parameter estimates from the general linear models. The step count composition alone explained 4.3% of the variance in BMI z-scores using least squares linear regression. After adjusting for the nested data structure and student race/ethnicity, a higher percentage of steps accrued during physical education, relative to other school segments, was associated with lower BMI z-scores (b = −0.34, 95% CI: −0.65–−0.03, *p* = 0.036). Likewise, a higher percentage of steps accrued during recess, relative to the other school segments, was significantly associated with lower BMI z-scores (b = −0.47, 95% CI: −0.83–−0.11, *p* = 0.012). Conversely, a higher percentage of steps accrued during lunchtime (b = 0.73, 95% CI: 0.24–1.22, *p* = 0.003), relative to the other school segments, was significantly associated with higher BMI z-scores. Steps accrued during academic class time relative to other segments was not significantly associated with BMI z-scores (*p* = 0.195).

Using the coefficients obtained from the general linear models, compositional isotemporal substitution determined how reallocation of accrued steps between school segments was associated with changes in BMI z-scores. Table 3 communicates the BMI z-score change score data in response to the reallocation of steps between each pair of school day segments. Academic class time data were not reported because of no statistical significance from the general linear model. A visual representation of the point estimate data provided in Table 3 is presented in Figure 1. The largest changes in BMI z-scores, in terms of lowering BMI z-scores, were observed when steps were reallocated from lunchtime to recess and from lunchtime to physical education (lower and right side of Figure 1). The largest increases in BMI z-scores were observed when steps were reallocated from recess to lunchtime and from physical education to lunchtime (upper and left side of Figure 1). Reallocating steps from recess to physical education, and vice versa, was associated with little change in BMI z-scores.

## 4. Discussion

The purpose of this study was to apply CoDA for the analysis of segmented school steps and associate the school step count composition to BMI z-scores in a sample of elementary school-aged children. The results indicated that a higher percentage of total school day steps accrued during physical education and recess, relative to other school segments, was associated with lower BMI z-scores. Conversely, a higher percentage of total school day steps accrued during lunchtime, relative to other segments, was associated with higher BMI z-scores. Reallocation of steps between specific pairs of school segmented was associated with potentially meaningful variation in BMI z-scores. Interpretation and discussion of these findings and implications for school physical activity programming are provided further.

The CoDA descriptive results suggest that steps accrued during lunchtime and physical education, recess and physical education, and lunchtime and recess tended to have a greater degree of co-dependency. These findings were evident from the compositional variation matrix and highlight that school segments that provide opportunities for physical activity tend to be co-dependent during the school day. For example, activity accrued during recess may influence activity accrued during physical education and lunchtime. Steps accrued during academic class time, however, were independent from the other school segments. Activity during class time may be low due to its academic nature and the seated sedentary behaviors that are often prevalent during academic learning time [29]. Using CoDA, we were able to quantify this independence of activity accrued during the academic class time compared to other school segments. Another descriptive finding was that boys accrued a lower percentage of school day steps during physical education compared to girls but accrued a higher percentage of school day steps during recess, lunchtime, and academic class time compared to girls (See Appendix A). This specific finding highlights the importance of structured physical activity for girls. Studies have shown that physical activity in girls is lower compared to boys for both school day and total day physical activity [30,31,32]. These differences are especially prevalent upon commencement of early adolescence, where physical, social, and emotional development shape physical activity behavior [33,34]. The social and emotional factors that influence physical activity may play a stronger role in girls [35,36]. Consequently, girls may respond differently to physical activity programming than boys. A positive social environment and structured single sex environments may promote greater physical activity engagement in girls [34,35].

A salient finding from CoDA was that a higher percentage of total school day steps accrued during physical education and recess, relative to other school segments, was associated with lower BMI z-scores. This suggests that the intensity of physical activity accrued during these segments could be higher, thus yielding greater physical activity energy expenditure. In fact, physical education has been shown to be an important source for higher intensity physical activity [37]. There is also evidence that regular physical education is associated with weight outcomes and meeting recommendations for both physical education and recess associates with healthier weight trajectories [38,39,40]. These findings may be because physical education is a venue where the least active children have consistent opportunities for participating in physical activity at higher intensities [37,41]. Furthermore, higher intensity physical activity is correlated with higher levels of cardiorespiratory endurance [42], which may also mitigate risk for overweight and obesity within the pediatric population [43] and could be a potential mechanism for the associations found in the current study. Importantly, physical activity habits partially developed through effective physical education can track into adulthood and reduce risk for non-communicated cardiovascular and metabolic disease [44].

Using compositional isotemporal substitution, an approximate 5% step reallocation from lunchtime to either physical education or recess was associated with lower BMI z-scores. Steps accrued while participating in higher physical activity at higher intensities during physical education and recess compared to lunchtime may explain this association. A greater reallocation of steps, especially at 15% between the aforementioned school segments, was associated with a more drastic reduction in BMI z-scores. These non-linear associations have also been observed in other pediatric populations when applying compositional isotemporal substitution to accelerometer time-use data [18,19,28]. Even though a greater than 15% reallocation between segments and its associated BMI z-score reduction may be unrealistic, the potential weight-related benefit of accruing more relative steps during physical education and recess is evident using the CoDA analytic approach when reallocating steps between 5% and 15% of the school total. Maintaining adequate school steps during these time segments is also important given that changes in time spent in moderate-to-vigorous physical activity decreases during these segments in late childhood through early adolescence [45]. The positive impact of targeting the physical education and recess environments during interventions may also have clinical meaning, especially in overweight and obese pediatric populations where these types of interventions may be most impactful [4].

The current study indicates that providing both physical education and recess opportunities for children can yield health benefits in children. Previous work has illustrated that children accumulate the most physical activity during school on days when they have physical education and multiple recess opportunities compared to days when they only have lunchtime physical activity opportunities [46]. Importantly, a second recess period can increase step counts by 20% on days when children do not have physical education [46]. Previous research has also indicated that small changes made within the environment, including increased focus on fitness and limiting class sizes, can help increase physical activity during physical education and adding some structure and coaching/modeling can also have a positive impact on physical activity during recess [47,48]. The current study’s findings should be considered along with this previous research indicating that small changes to these segments can increase activity opportunities to positively impact obesity risk in primary school-aged children.

Strengths of the study include the analysis of segmented school day steps within the framework of CoDA. Pedometers are also an accessible and objective instrument to measure steps and a novel methodology employed within the CoDA literature. Limitations include use of a convenience sample of fourth and fifth grade children; therefore, the results do not generalize to younger or older age groups. The cross-sectional research design precludes causal inferences, and the true directionality of the observed associations cannot be ascertained. Additionally, in between school segments, there are brief transition periods where children are transported between different classes or between an academic class and physical education, recess, or lunch. This was not specifically accounted for within the analysis but was incorporated into the class time count data. However, it is assumed to have made little impact on the results given the brief nature of these transitions. Sociometric information such as socioeconomic status was not collected and may be a confounding variable in the observed associations. Finally, body mass index was used as an assessment of body composition; however, other anthropometric measurements such as waist circumference or a direct assessment of body fat such as skinfolds thickness may have provided greater construct validity.

## 5. Conclusions

The Institute of Medicine has recommended that children accumulate at least 30 min of moderate-to-vigorous physical activity during school hours [49]. Using receiver operating characteristic curves, it has previously been shown that accumulating approximately 5500 steps during school hours is associated with meeting this guideline with reasonable accuracy [50]. The findings from the current study provide important insight that perhaps the accumulation of the majority of school steps during physical education and recess may improve the accuracy of this step count recommendation because of the accumulation of higher intensity physical activity, which may favorably contribute to energy balance. When planning school-based physical activity interventions, targeting physical education and recess may have merit if outcomes are weight-related. The current study has shown the potential health-related benefit of accruing physical activity during physical education and recess relative to other school segments in children.

## Figures and Tables

**Figure 1 ijerph-18-03243-f001:**
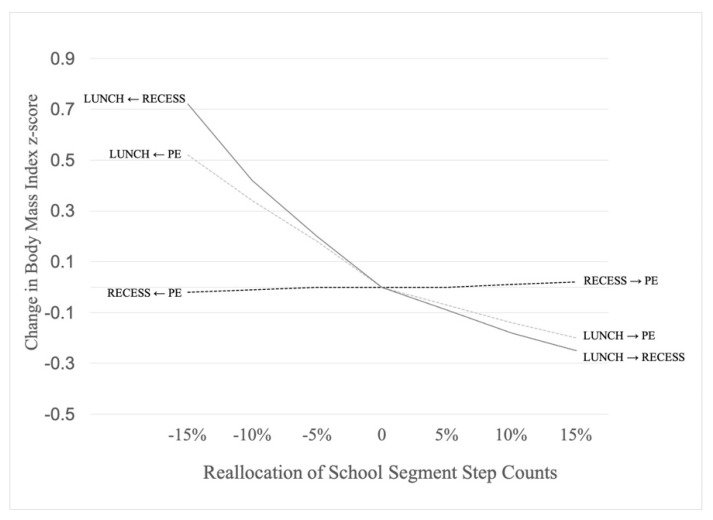
Change in body mass index z-scores per reallocation of step counts between school segments across a 7-h school day. Note: PE is step counts during physical education; RECESS is step counts during recess; LUNCH is step counts during lunch period.

**Table 1 ijerph-18-03243-t001:** Descriptive statistics for the total sample and within sex-specific groups.

Variable	Total Sample (Means and SD) (N = 855)	Girls (Means and SD) (n = 443)	Boys (Means and SD) (n = 412)
Age (years)	9.7 (1.0)	9.7 (1.0)	9.8 (1.0)
BMI (kg/m^2^)	19.4 (4.2)	19.1 (4.1)	19.6 (4.3)
BMI z-score	0.63 (1.11)	0.53 (1.11)	**0.73 * (1.10)**
Healthy Weight	521 (60.9%)	286 (64.6%)	235 (56.8%)
Overweight/Obese	334 (39.1%)	157 (35.4%)	177 (43.2%)

Note: BMI stands for Body Mass Index; bold and * indicates statistical differences between sexes, *p* < 0.05.

**Table 2 ijerph-18-03243-t002:** Parameter estimates from the Body Mass Index z-score naïve and general linear mixed effects regression models using compositional data analysis.

Isometric Log Ratio Predictor	Unadjusted Model b-Coefficient (95% CI)	Unadjusted Model *p*-Value	Adjusted Model ^†^ b-Coefficient (95% CI)	Adjusted Model ^†^ *p*-Value
*ilr* _PE/RECESS × LUNCH × CLASS_	**−0.44 * (−0.72, −0.16)**	0.002	**−0.34 * (−0.65, −0.03)**	0.036
*ilr* _RECESS/PE × LUNCH × CLASS_	**−0.40 * (−0.76, −0.04)**	0.029	**−0.47 * (−0.83, −0.11)**	0.012
*ilr* _LUNCH/PE × RECESS × CLASS_	**0.74 * (0.25, 1.22)**	0.003	**0.73 * (0.24, 1.22)**	0.003
*ilr* _CLASS/PE × RECESS × LUNCH_	−0.09 (−0.02, 0.21)	0.111	−0.08 (−0.04, 0.20)	0.195

Note: *ilr* stands for isometric log ratio; 95% CI stands for 95% Confidence Interval; ^†^ adjusted model accounts for school and classroom level clustering and student race/ethnicity; PE is step counts during physical education; RECESS is step counts during recess; LUNCH is step counts during lunch period; CLASS is step counts within the academic classroom; outcome is body mass index z-scores; bold and * denotes statistical significance, *p* < 0.05.

**Table 3 ijerph-18-03243-t003:** Predicted changes in Body Mass Index z-scores following step count reallocation between school day segments using compositional isotemporal substitution.

Reallocation	Δ BMI z-Score 250 Step Reallocation (~5% of Total)	Δ BMI z-Score 500 Step Reallocation (~10% of Total)	Δ BMI z-Score 750 Step Reallocation (~15% of Total)
RECESS to PE	0.00	0.01	0.02
LUNCH to PE	−0.07	−0.14	−0.20
LUNCH to RECESS	−0.09	−0.18	−0.25
PE to RECESS	0.00	−0.01	−0.02
PE to LUNCH	0.18	0.34	0.52
RECESS to LUNCH	0.20	0.42	0.72

Note: PE is step counts during physical education; RECESS is step counts during recess; LUNCH is step counts during lunch period; Academic classroom step count data not reported because of no statistical significance.

## Data Availability

The data presented in this study are available on request from the corresponding author.

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
