# Peer review of "Segmented School Physical Activity and Weight Status in Children: Application of Compositional Data Analysis"

_ijerph, 2021, doi:10.3390/ijerph18063243_

Round 1
Reviewer 1 Report
The introduction presents little background about the problem addressed in the article, it only presents a single perspective (line 36). Not enough conceptual basis is provided on the causes of the problem or on similar experiences or studies.
It presents a description of the basic analysis technique that serves as the basis for the article, but a minimal description of its foundations is not explained.
A sociometric description of the participants would have been interesting and convenient not only the information of the webpage of school.
The content of the classes and the activities carried out in the other spaces in which the children's activity was controlled have not been described. The type, quantity and quality of the tasks carried out in the classes is a determining factor for the study.
There is no justification for why age and sex were not taken into account but race was taken into account in the analysis.
In the statistical analysis the explanation of the permutations does not appear, it would be convenient to do it.
Some parts of the discussion are not obtained directly from the results obtained (lines 260-265). The same can be said of any conclusion (lines 317-320)
Author Response
Reviewer #1:
The introduction presents little background about the problem addressed in the article, it only presents a single perspective (line 36). Not enough conceptual basis is provided on the causes of the problem or on similar experiences or studies.
-Thank you for this comment. We have now added additional perspectives within the Introduction section and further develop the research problem (lines 42-48, 78-83).
It presents a description of the basic analysis technique that serves as the basis for the article, but a minimal description of its foundations is not explained.
-Thank you for this comment. This additional background information has now been provided within the Introduction section (lines 56-59).
A sociometric description of the participants would have been interesting and convenient not only the information of the webpage of school.
-Thank you. We agree with this comment. Unfortunately, we did not have access to this information when we collected the data from the children. We now mention this as a limitation within the Discussion section (line 342-343).
The content of the classes and the activities carried out in the other spaces in which the children's activity was controlled have not been described. The type, quantity and quality of the tasks carried out in the classes is a determining factor for the study.
-Thank you. These activities have now been provided within the Methods section (lines 124-131).
There is no justification for why age and sex were not taken into account but race was taken into account in the analysis.
-Thank you. Age and sex were used to calculate BMI z-scores. Because of this, we believe that their inclusion in the model would be redundant (lines 176-178).
In the statistical analysis the explanation of the permutations does not appear, it would be convenient to do it.
-Thank you for this comment. We now explain sequential permutation within the statistical analysis section (line 164-172).
Some parts of the discussion are not obtained directly from the results obtained (lines 260-265). The same can be said of any conclusion (lines 317-320).
-Thank you for this comment. We believe the prior content could be a mechanism for the observed associations; we now comment on this within the Discussion (lines 298-299). The Conclusion has been amended accordingly (lines 354-361).
Reviewer 2 Report
First, I would like to congratulate the authors on an overall excellent manuscript. The amount of work that went into collecting this large sample size is appreciated. I believe this information is novel and impactful and may help to direct decisions on PA reform in schools. Many of the edits I have to offer are just stylistic which are not worth changing because they do not technically detract from the manuscript. So, I have only minor comments:
For the keywords, authors should only include terms that are not contained in the title. This will allow others to discover the manuscript easier. Perhaps replace the word “children” with “body mass index”
The intro is concise and well written. However, I think a large part of the research problem is missing from the intro which is how physical education and physical activity are largely becoming designed out of the school day. This discussion may be even more important given that the two segments which appear to be most affected currently (i.e. recess and PE) are the very activities being threatened. Just a suggestion.
Line 78: Was there a specific reasoning for 4th and 5th graders other than convenience?
Line 94: Since these measurements were taken during PE, I assume kids were wearing light gym clothing that would add minimal weight to the measurement?
The percentages in the table do not add up to 100% for some of the values. This is likely just due to rounding, but authors should amend.
Line 242: “smaller percentage”
Line 264: A sentence on how developing physical activity habits early is linked to better PA patterns later in life would be beneficial here.
A limitation which should be added is the use of BMI. While more convenient and easier to measure, waist circumference may be a better predictor of weight-associated morbidity. Among other measurements.
Author Response
Reviewer #2:
For the keywords, authors should only include terms that are not contained in the title. This will allow others to discover the manuscript easier. Perhaps replace the word “children” with “body mass index”
-Thank you. The keywords have been amended as suggested (line 22).
The intro is concise and well written. However, I think a large part of the research problem is missing from the intro which is how physical education and physical activity are largely becoming designed out of the school day. This discussion may be even more important given that the two segments which appear to be most affected currently (i.e. recess and PE) are the very activities being threatened. Just a suggestion.
-Thank you for this important comment. The research problem has now been expanded upon at the end of the Introduction section (lines 78-83).
Line 78: Was there a specific reasoning for 4th and 5th graders other than convenience?
-Thank you. Data collection was not permissible for younger grade levels - this has now been indicated in the Methods section (lines 101-102).
Line 94: Since these measurements were taken during PE, I assume kids were wearing light gym clothing that would add minimal weight to the measurement?
-Thank you. The children were wearing light clothing and had their shoes off during the weight measurement. This has been added to the Methods section (lines 111-112).
The percentages in the table do not add up to 100% for some of the values. This is likely just due to rounding, but authors should amend.
-Thank you, changes have now been made within Table 1.
Line 242: “smaller percentage”
-Thank you, this word has been changed (line 274).
Line 264: A sentence on how developing physical activity habits early is linked to better PA patterns later in life would be beneficial here.
-Thank you. This important comment has now been made within the Discussions section (lines 299-301).
A limitation which should be added is the use of BMI. While more convenient and easier to measure, waist circumference may be a better predictor of weight-associated morbidity. Among other measurements.
-Thank you. This has now been added to the limitations paragraph within the Discussion section (lines 343-346).